

# Sex ratio and relatedness in the Griffon vulture (*Gyps fulvus*) population of Serbia

Slobodan Davidović[1,2], Saša Marinković[2,3], Irena Hribšek[2,4], Aleksandra Patenković[1], Marina Stamenković-Radak[1,5] and Marija Tanasković[1]

[1] Department of Genetics of Populations and Ecogenotoxicology, Institute for Biological Research "Siniša Stanković"—National Institute of the Republic of Serbia, University of Belgrade, Belgrade, Serbia
[2] Birds of Prey Protection Foundation, Belgrade, Serbia
[3] Department of Ecology, Institute for Biological Research "Siniša Stanković"—National Institute of the Republic of Serbia, University of Belgrade, Belgrade, Serbia
[4] Natural History Museum Belgrade, Belgrade, Serbia
[5] Faculty of Biology, University of Belgrade, Belgrade, Serbia

Corresponding author
Slobodan Davidović,
slobodan.davidovic@ibiss.bg.ac.rs

## ABSTRACT

**Background:** Once a widespread species across the region of Southeast Europe, the Griffon vulture is now confined to small and isolated populations across the Balkan Peninsula. The population from Serbia represents its biggest and most viable population that can serve as an important reservoir of genetic diversity from which the birds can be used for the region's reintroduction programmes. The available genetic data for this valuable population are scarce and as a protected species that belongs to the highly endangered vulture group, it needs to be well described so that it can be properly managed and used as a restocking population. Considering the serious recent bottleneck event that the Griffon vulture population from Serbia experienced we estimated the overall relatedness among the birds from this population. Sex ratio, another important parameter that shows the vitality and strength of the population was evaluated as well.

**Methods:** During the annual monitoring that was performed in the period from 2013–2021, we collected blood samples from individual birds that were marked in the nests. In total, 169 samples were collected and each was used for molecular sexing while 58 presumably unrelated birds from different nests were used for inbreeding and relatedness analyses. The relatedness was estimated using both biparentally (10 microsatellite loci) and uniparentally (*Cytb* and *D-loop I* of mitochondrial DNA) inherited markers.

**Results:** The level of inbreeding was relatively high and on average it was 8.3% while the mean number of relatives for each bird was close to three. The sex ratio was close to 1:1 and for the analysed period of 9 years, it didn't demonstrate a statistically significant deviation from the expected ratio of 1:1, suggesting that this is a stable and healthy population. Our data suggest that, even though a relatively high level of inbreeding can be detected among the individual birds, the Griffon vulture population from Serbia can be used as a source population for restocking and reintroduction programmes in the region. These data combined with previously observed genetic differentiation between the populations from the Iberian and Balkan Peninsulas suggest that the introduction of foreign birds should be avoided and that local birds should be used instead.

# INTRODUCTION

The Eurasian Griffon vulture (*Gyps fulvus*) is a colonial, cliff-nesting obligate scavenger specialized in feeding on softer parts of mammal carcasses (*Johnson et al., 2006*). As a member of a vulture group of birds, it provides an essential ecosystem service of carcass removal in nature which prevents the spread of infectious diseases (*Moleon et al., 2014*). They are monogamous, long-living birds with delayed maturity and a maximum of one fledgling per pair per year (*Del Hoyo, Elliott & Sargatal, 1994*; *Fergusson-Lees & Christie, 2001*; *García-Ripollés & López-López, 2011*; *Pirastru et al., 2021*). Although once common species in Europe, these vultures faced a serious population decline in the mid-20th century leading to the extinction of many populations across the European range and the vast depletion of breeding birds across the breeding range (*Martens, Kelemen & Hodor, 2005*; *Pantovic & Andevski, 2018*; *Pirastru et al., 2021*; *Schenk, 1972*; *Tucker & Heath, 1994*). Thanks to the successful cross-continental conservation efforts that included reintroduction and conservation measures implemented across the former natural range of the Griffon vulture in Europe, the future of this ecologically important species seems brighter (*Dobrev et al., 2021*; *Le Gouar et al., 2008*; *Pirastru et al., 2021*; *Potena et al., 2009*). The European populations are now showing a significant increment in size with this species now being categorized as 'Least Concern' on the IUCN Red List (*BirdLife International, 2021*). However, even with these successful measures, this species is still facing serious population declines in some habitats and has become extinct in others, which is why it is a subject of conservation strategies by numerous laws, directives, and conventions (reviewed in *Pirastru et al., 2021*). Due to the bottlenecks and inbreeding depressions that most of these populations experienced in recent history, constant monitoring efforts are needed in order to assess the population's ability to maintain healthy status and possibly be future reintroduction resources.

Due to conservation efforts that included the opening of the feeding station and prohibition of poison usage, the Serbian Griffon vulture population is one of the successful conservation stories. From only ten breeding pairs recorded in the 1990s, it has reached 164 breeding pairs in 2019 and now is the largest population in the Balkan Peninsula (*Dobrev et al., 2021*; *Marinkovic et al., 2021*). The historical bottleneck, seen in the last decade of the 20th century, was a peak of a rapid demographic decline in Serbian Griffon vulture populations. This decline resulted from mass poisonings and loss of food sources due to the change in livestock management (*Davidovic et al., 2020*). A study by *Davidovic et al. (2020)* showed that this population's genetic variability (based on the variability of 10 microsatellites loci) was similar to numerous, unscathed populations such as the Iberian Griffon vulture population and therefore is proposed to serve as a stock population for further reintroduction efforts in the Balkan Peninsula. In addition, analysis of the mitochondrial DNA (mtDNA) gene, *Cytb*, further demonstrated that the Griffon vulture population of Serbia exhibits substantial levels of genetic diversity as well as the presence of
private haplotypes which stresses the need of using the local birds in the future reintroduction programmes planned for the region (*Davidovic et al., 2022*). However, determining the status of the endangered population under conservation measures should be based on more than just 1 or 2 parameters of genetic variability. Parameters such as breeding parameters, genetic analysis of mitochondrial genome, adult and offspring sex ratio as well as relatedness among presumed unrelated birds are of great importance for establishing the status of both biological and genetic health of the population in question (*Arshad, Chaudhary & Wink, 2009*; *Arshad et al., 2009*; *Earnhardt, 1999*; *Friar et al., 2001*; *Godoy et al., 2004*).

Breeding parameters in the Serbian Griffon vulture population are relatively high despite the substantial increase in population size, implying that the population is far from its saturation level (*Marinkovic et al., 2021*) further strengthening this population prospect as a reintroduction resource in the Balkan Peninsula (*Davidovic et al., 2020*, *2022*). The latest research on the spatial ecology of Griffon vultures in the Balkans identified seven key vulture zones (one in Serbia, one shared between North Macedonia and Bulgaria, one shared between Bulgaria and Greece, two in Bulgaria, one in western Greece and one shared between Kvarner Archipelago islands in Croatia and the Julian Alps—Italy, Austria and Slovenia) (*Peshev et al., 2021*). This research also identified that the young birds tend to migrate but always return to their nesting ground after migration due to the strong natal philopatry specific for this species and so far no or little exchange of migrant birds between these vulture zones in the Balkans was detected (*Peshev et al., 2021*).

Sex ratio, the proportion of females and males, is an important factor contributing to the success of the newly established population (*Sarrazin & Legendre, 2000*). *Fisher (1930)* argument that equal investment in the sex of offspring represents an evolutionary stable strategy and *Trivers' (1972)* additional comments in theory on sexual selection states that at birth in monogamous species expected sex ratio should be 1:1 and that the equal ratio may be a useful indicator of population trajectory and overall health. In sexually monomorphic species, where there are no visible and easily detectable morphological differences between sexes, the importance of monitoring and maintaining an equal sex ratio grow stronger especially if the population is small and endangered or is under consideration to be used as a reintroduction resource (*Legendre, 2004*). The success of reintroduction projects heavily depends on the accurate determination of the sex of founding individuals (*Griffiths & Tiwari, 1995*; *IUCN, 1998*). In such species, the most common strategy is a random sampling of individuals hoping for a balanced sex ratio in the founding group and this strategy heavily relies on the implication that the sex ratio is equal at birth and that mortality and dispersal of adults are equal between sexes (*Bosé et al., 2007*). Stochastic fluctuations of sex ratio in small populations of monogamous species could substantially reduce effective population size by increasing reproductive variance among individuals as well as increasing extinction probability (*Garofalo et al., 2016*; *Legendre et al., 1999*). Therefore, an equal sex ratio is one of the factors contributing to the maximum population growth rate (*Legendre, 2004*). On the other hand, the adult sex ratio, especially in monogamous species, can sometimes be skewed toward males since different environmental pressures affects opposite sexes differently. Traditional sex determination

of individual adult Griffon vulture is difficult, as is the case with other monomorphic species, and requires highly invasive methods like cloacal examination (*Tacha & Lewis, 1979*) and laparoscopy (*Fry, 1983*). Thus, molecular sexing was developed to replace these invasive and difficult methods for fieldwork (*Fridolfsson & Ellegren, 1999*). There are two possible ways of acquiring samples for molecular sexing: capturing adult birds or sampling chicks in the nests. Considering the difficulties of capturing adult birds as well as the physical stress they experience in the process it is more efficient and humane to sample birds while they are in the nests (*Beja-Pereira et al., 2009*). This sampling strategy initially provides only the information about the offspring sex ratio, but thanks to the marking of the chicks, these birds can later be monitored as adults thus enabling information about the adult sex ratio. On the other hand, it is important to know the sex ratio of the hatched birds because the sex ratio among adults can be influenced by external factors like different life traits or deaths caused by accidents that can randomly change the sex ratio among adult birds.

The Serbian Griffon vulture population experienced a similar fate as other populations in Europe in the form of serious population decline. Although it made it prone to drift and bottleneck effects this population retained significant genetic variability similar to those detected in stable and large populations (*Davidovic et al., 2020*; *Davidovic et al., 2022*). Nevertheless, there is always a concern about the relatedness and kinship of remaining birds due to their natal philopatric behaviour and inbreeding imprint on the population's overall fitness. For genetic-based estimates of relatedness, in populations with no available pedigree data, traditional markers of choice are microsatellites, usually a panel of eight—30 microsatellite loci (*Attard, Beheregaray & Möller, 2018*; *Galla et al., 2020*; *Jones et al., 2010*; *Pemberton, 2008*; *Taylor, 2015*; *Wang, 2011*). Traditionally, the relatedness of two individuals is the probability that two randomly chosen alleles are identical by descent (IBD) (*Speed & Balding, 2015*). Currently, available relatedness estimation methods can produce varying results since each takes into account different assumptions about the input data (*Hogg et al., 2018*). The program COANCESTRY (*Wang, 2011*) reflects population reality because it takes inbreeding into account. It is especially useful in inferring relationships in populations with historically recorded bottleneck events. COANCESTRY estimates two types of identification of related individuals. The first estimate is relatedness ($r$) which is a continuous measure of the proportions of the alleles of the dyad (pair of individuals) that are IBD relative to a reference population. The second type infers kinship which are discrete categories of dyad relationships (parent-offspring, full-sibling, half-sibling). Additionally, an inbreeding coefficient can be calculated for an individual as well as for the population in question. Inbreeding coefficients may serve as predictors of the population's overall fitness and molecular-based pedigrees may be useful in the management and further reintroduction efforts enabling the choice of unrelated founder individuals (*Galla et al., 2020*).

In this article, we evaluated the sex ratio and estimated the relatedness among Griffon vulture offspring hatched in Serbia. The aims of this study were to first assess the impact of the bottleneck event that occurred in the early 1990s and second to further investigate the

potential of using this population as a stock population for reintroduction in the Balkan Peninsula.

## MATERIALS AND METHODS

### Population location and sampling

The sampling of the Griffon vulture population that inhabits the gorge of the river Uvac in the Southwestern part of Serbia was performed in the period from 2013–2021. The sampling was conducted directly in the nests as described in *Davidovic et al. (2020)*. Blood samples were collected *via* wing veni puncture using sterile equipment with respect to animal welfare following relevant guidelines and regulations. During the sampling, a thorough morphological analysis of chicks was performed (weight, bird length, wing length, tail length, hindquarters and finger length, tarsus length, beak length, width and height) and they were tagged by the licensed personnel. None of the birds were sacrificed or injured during the sampling process and all the birds were immediately returned to their original nests, or in the case of the captured/recaptured subadult birds, released to nature. In total, we have collected the blood samples from 169 young birds that were stored either in sodium citrate or QUEENS buffer at −20 °C. In among 169 blood samples taken from the birds, four belonged to the birds that were recaptured in 2013 but were previously marked in the year of their hatching 2011 (one bird) and 2009 (three birds). All the procedures applied in this study were reviewed and approved by the Ministry of Nature Protection of the Republic of Serbia, the Ministry of Agriculture, Forestry and Water Management of the Republic of Serbia (323-07-09135/2020-05/1) and the ethical committee of the Institute for Biological Research "Siniša Stanković", Belgrade (323-09135-2020-05).

### DNA extraction

The DNA extraction was performed by using the GeneJET Genomic DNA Purification Kit (Cat. No. K0721; Thermo Fisher Scientific, Waltham, MA, USA) and a modified salting-out protocol as described in *Davidovic et al. (2022)*. The quality of the DNA extracts was checked both by the spectrophotometer (NanoPhotometer, IMPLEN, Westlake Village, CA, USA) and agarose gel electrophoresis.

### Molecular sexing

Molecular sexing of all sampled birds was performed using the protocol described in *Chang et al. (2008)*. Fragments for the Chromo-Helicase-DNA binding protein (*CHD*) located on chromosomes Z (*CHD-Z*) and W (*CHD-W*) were amplified in two separate reactions using the following set of primers: P2 reverse primer 5′ TCTGCATCGCTAAAT CCTTT 3′ complementary for both W and Z chromosomes, and two forward primers *CHD-ZW*-common 5′ GATCAGCTTTAATGGAAGTGAAG 3′ and *CHD-W*-specific 5′ GGTTTTCACACATGGCACA 3′. Amplification was performed in a volume of 25 μL with the following final concentrations of reaction components: 1×Taq Buffer with $(NH_4)_2SO_4$, 2.5 mM $MgCl_2$, 0.8 mM dNTP mix, 1 U of reverse Taq polymerase

(all components were produced by Thermo Fisher Scientific, Waltham, MA, USA) and 10 pmol of each forward and reverse primer. All PCRs were performed using the following program: one cycle of initial denaturation at 94 °C for 5 min, after which there were 38 cycles of 35 s at 94 °C, 35 s at the annealing temperature at 56 °C and 35 s at 72 °C. The step of final elongation was performed at 72 °C for 10 min. All amplified fragments were evaluated by agarose gel electrophoresis. The lengths of the PCR fragments were as follows: 148 bp for *P2/CHD-ZW*-common set of primers and 258 bp for *P2/CHD-W*-specific set of primers. Both PCR fragments (148 and 258 bp) were detected for the heterogametic female birds and for the homogametic male birds only one fragment (148 bp) was detected. Fragments of interest were amplified in a 2720 Thermal Cycler (Applied Biosystems, Waltham, MA, USA). A portion of the total analysed birds (54 birds) whose sex was previously determined for the period of 2013–2018 were taken from the *Hribsek (2022)*.

## Sequencing

Sequencing of mtDNA regions *Cytb* and *D-loop I* were performed for the chosen birds in order to determine their maternal relationship. The chosen fragments were amplified with the following set of primers: for *Cytb* forward GF-L13740 5′ TAATCAACAACTCCCTA ATCGACCTAC 3′, reverse GF-H15014 5′ CCTTTTGGGCCGAGAACTCT 3′ and for *D-loop I* forward GF-L14961 5′ GCTCAACCACTAAATACTCTAATAG 3′, reverse GF-H16307 5′ CAGTTTAGGGGGGAAGGAAGG 3′.

The same PCR program was used for the amplification of both *Cytb* and *D-loop I* fragments: one cycle of initial denaturation at 94 °C for 5 min, after which there were 35 cycles of 35 s at 94 °C, 35 s at the annealing temperature at 55 °C and 90 s at 72 °C. The step of final elongation was performed at 72 °C for 10 min. The lengths of the PCR fragments were as follows: 1,320 bp for the GF-L13740/GF-H15014 for *Cytb* set of primers and 1,391 bp for the GF-L14961/GF-H16307 set of primers for *D-loop I*. Amplification of the mtDNA regions was performed using the same reaction components as the ones used for the PCR amplification of *CHD* genes. The sequencing was performed by Macrogen Europe using the same set of primers that were used for the PCR amplification of *Cytb* while the *D-loop I* region was sequenced with a different pair of primers in order to obtain a better quality sequence and avoid the T and C tract (GF-L15147 5′ TGTACATTACAC TATTTGCCCCATA 3′ and GF-H15547 5′ GCAGGGGGGAAAGTAAGATCC 3′). Mitochondrial haplotypes were determined by comparison with the mtDNA reference sequence for Griffon vulture, NC_036050.1 (*Mereu et al., 2017*), following the nomenclature that includes the nucleotide position and adequate substitution different to the reference mitogenome. *Cytb* sequences used in this analysis were published in *Davidovic et al. (2022)* and can be accessed by the following accession numbers: OL962644, OL962685, OL962664, OL962652, OL962667, OL962636, OL962648, OL962634, OL962646 and OL962649. Newly sequenced *D-loop I* sequences were deposited in GenBank under the following accession numbers: ON988042–ON988051.

## Data analysis

For the estimation of relatedness, $r$, among the pair of birds (dyads) we used 57 presumably unrelated birds originating from 56 different nests from the total collected sample of 169 birds. In this analysis we used the published genetic data of ten microsatellite loci (*Davidovic et al., 2020*). In order to test the results of relatedness estimates, we used two birds from the same nest (as known full-siblings, marked S95 and S53) thus having a total of 57 individuals in our analysis. The estimation was performed using the COANCESTRY program (*Wang, 2011*). All seven relatedness estimators (TrioMl, (*Wang, 2007*); Wang, (*Wang, 2002*); LynchLi, (*Li, Weeks & Chakravarti, 1993*; *Lynch, 1988*); LynchRd, (*Lynch & Ritland, 1999*); Ritland, (*Ritland, 1996*); QuellerGt, (*Queller & Goodnight, 1989*); DyadMl, (*Milligan, 2003*)) including three inbreeding estimators (TrioMl, (*Wang, 2007*); Ritland, (*Ritland, 1996*); DyadMl, (*Milligan, 2003*)) implemented in the COANCESTRY program were tested in this analysis. Bootstrapping of 10,000 samples over loci were performed in order to obtain the 95% confidence interval (95% CI) of each estimator for each dyad. Using the COANCESTRY program with the same parameter settings, as previously defined, we evaluated the inbreeding coefficients, $F$, for each analysed bird. The threshold values of $r_{xy}$ coefficient were as follows: for $r_{xy} < 0.09375$ the birds in the dyad were unrelated, $r_{xy} < 0.1875$ and $\geq 0.09375$ were considered as third-order relatives (relationship), $r_{xy}$ values $< 0.375$ and $\geq 0.1875$ were considered as second-order relatives (relationship) and $r_{xy}$ values $\geq 0.375$ were considered as having full-sibling or parent-offspring relationship (*del Pazo et al., 2021*; *Lynch & Ritland, 1999*). In order to confirm acquired parent-offspring, full-siblings and half-siblings relationships additional analysis in the software package ML-Relate (*Kalinowski, Wagner & Taper, 2006*) was performed.

Pearson's $\chi^2$ goodness-of-fit statistics (*Wilson & Hardy, 2002*) was used to test the statistical significance for departure from an expected sex ratio of 0.5.

## RESULTS

### Sex ratio

The results of the molecular sexing for the period of 2013–2021 are presented in Table 1. The number of hatched chicks per year was in constant increase with the exception of the years 2014 and 2021. The number of sampled chicks per year varied from 13 to 29 which heavily depended on the weather conditions during the sampling process as well as the availability of the nests that are mostly located on cliffs with difficult access. Some of the nests were visited multiple times during this period. The chicks originating from the supposed same breeding pair of birds that were hatched in different years were sampled. The results of molecular sexing for the chicks originating from the same nests are presented in Table S1. These data show the gender alteration in the nest by year. In some of the nests, one of the sexes was more prominent than the other while there are also nests with equal distribution of sexes throughout the sampling period. During the period of 9 years, we were able to sample between 12.04–17.68% of all hatched chicks per year and the only year with low sampling success was 2019 when only 8.97% of the hatched chicks

**Table 1 The ratio of 1,182 hatched and 169 sampled chicks of the Griffon vulture population from Serbia with the sex ratio for the period of 2013–2021.**

| Year | Nh | Ns | Nm | Nf | Rs (%) | Rm (%) | Rf (%) |
|------|------|------|------|------|--------|--------|--------|
| 2013 | 112 | 16 | 10 | 6 | 14.29 | 62.50 | 37.50 |
| 2014 | 108 | 13 | 7 | 6 | 12.04 | 53.85 | 46.15 |
| 2015 | 113 | 16 | 7 | 9 | 14.16 | 43.75 | 56.25 |
| 2016 | 119 | 21 | 11 | 10 | 17.65 | 52.38 | 47.62 |
| 2017 | 125 | 17 | 9 | 8 | 13.60 | 52.94 | 47.06 |
| 2018 | 144 | 20 | 9 | 11 | 13.89 | 45.00 | 55.00 |
| 2019 | 145 | 13 | 6 | 7 | 8.97 | 46.15 | 53.85 |
| 2020 | 164 | 29 | 11 | 18 | 17.68 | 37.93 | 62.07 |
| 2021 | 148 | 20 | 10 | 10 | 13.51 | 50.00 | 50.00 |
| 2013*–2021 | 1,182# | 169 | 81 | 88 | 14.29 | 47.93 | 52.07 |

Notes:
2013*, Including the birds captured in 2009 (three female birds) and 2011 (one male bird).
#Total number of hatched chicks for the period 2013–2021 including the birds hatched in 2009 and 2011.
Nh, total number of hatched chicks detected in the nests for a given year; Ns, number of sampled birds; Nm, number of males; Nf, number of females; Rs, a percentage of sampled chicks in relation to the total number of hatched chicks for the given year; Rm, a ratio of male chicks in a sample for a given year; Rf, a ratio of female chicks in a sample for a given year.

were sampled mainly because of the bad weather conditions (Table 1). Due to the incident in which a number of young birds fell from the nests into the lake, the sampling success of the hatched chicks in 2020 was somewhat higher compared to previous years. Four birds that fell from the nests were rescued, marked and their blood samples were taken for genetic analyses.

The sex ratio among hatched chicks for each year is presented in Table 1. There was no statistically significant deviation from the expected sex ratio of 0.5 (Table 2) although, somewhat skewed results were observed in 2013 (62.50% males towards 37.50% females) and 2020 (37.93% males towards 62.07% females) (Table 1). In addition, the sex was determined for the four birds recaptured in 2013, although they were first captured and marked in 2009 (three females S033, S052, BE423) and in 2011 (male S078). These birds were not included in the statistics for the year 2013 since they were born in different years before the blood sampling from the fledglings became a part of the regular monitoring. No statistically significant departure from the expected sex ratio was observed when all sexed individuals in the period 2013–2021 were compared (Table 2) even though a slightly higher proportion of the female fledglings can be observed (Table 1).

## Relatedness

Relatedness was estimated for the 57 birds originating from 56 different nests. According to the reproductive behaviour of Griffon vultures, birds originating from the different nests should not be closely related which is why these 57 birds were chosen for the relatedness analysis. During a lifetime, one pair of Griffon vultures uses the same nest throughout their reproductive period so there should be no siblings among chicks originating from different nests. The genetic data used for the estimation of relatedness were obtained from the

**Table 2 Statistical significance of balance between the sexes for each year in the period of 2013–2021 for the sampled chicks of the Griffon vulture population from Serbia.**

| Year | Nh | $p$ |
|---|---|---|
| 2013 | 112 | 0.80259 |
| 2014 | 108 | 0.93868 |
| 2015 | 113 | 0.90052 |
| 2016 | 119 | 0.96202 |
| 2017 | 125 | 0.95309 |
| 2018 | 144 | 0.92034 |
| 2019 | 145 | 0.93868 |
| 2020 | 164 | 0.80926 |
| 2021 | 148 | 1.00000 |
| 2013[*]–2021 | 1,182[#] | 0.96696 |

Notes:
2013[*], Including the birds captured in 2009 and 2011.
Nh, total number of hatched chicks detected in the nests for a given year; $p$, statistical significance $p$-value
Parameters used for the analysis: $\chi^2 = 3.841$, df = 1.

analysis of 10 microsatellite loci analysed in the previous work of *Davidovic et al. (2020)* and presented in Table S2.

The relatedness was calculated using all seven available estimators in the COANCESTRY software. Of all estimators, the TrioML estimator (*Wang, 2007*) exhibited the lowest variance and it was chosen to represent the relatedness among the pair of birds (dyads) in the analysed sample (Tables S3 and 3) (*Wang, 2011*). TrioML uses a likelihood method that estimates the IBD coefficients using the multilocus genotypes of the focal dyad as well as a reference individual (*Wang, 2007, 2011*). In the same way, this estimator can calculate individual inbreeding coefficients using the estimates of the nine IBD coefficients (*Wang, 2007, 2011*). The mean value of the relatedness $r_{xy}$ among the sampled birds together with the acquired variance is presented in Table 3. According to the TrioML estimator, each bird from the analysed sample of the Griffon vulture population from Serbia on average has almost three relatives of the full-sibling or parent-offspring relationship, almost eight half-siblings or second-order relatives and around nine third-order relatives (Tables 3 and S3). When 95% CI for TrioML estimator is taken into account, the number of relatives per bird is lower, around one bird of second-order relationship and two birds of third-order relationship while the number of birds with full-sibling or parent-offspring relationship is rather low and close to zero (Tables 3 and S4). Overall, each bird in the sample has around 20 relatives according to the TrioML estimator and around three relatives according to 95% CI for the same estimator (Tables 3 and S3–S4). The complete list of mean relatedness for each estimator with appropriate variance and correlation between estimators is presented in Table S5. The number of relatives that each individual bird has according to the TrioML estimator is presented in Table S6. Inbreeding coefficients, $F$, for each bird used in this analysis are presented in Table S7. According to the TrioML estimator which exhibits the lowest variance the mean $F = 0.08268$ (Table S7).

**Table 3 Relatedness estimates based on the TrioML estimator for individual pairs of diploid individuals and average numbers of birds that have full-siblings, second-order and third-order relationships for each bird in the sample of the Griffon vulture population.**

|  | Relatedness (estimate) | Relatedness (95% CI) |
|---|---|---|
| $r_{xy}$ | 0.09669 | 0.01334 |
| Variance | 0.01701 | 0.00234 |
| Afs | 2.98 ± 2.21 | 0.17 ± 0.38 |
| Asor | 7.93 ± 3.14 | 0.95 ± 1.02 |
| Ator | 9.47 ± 2.74 | 1.86 ± 1.39 |
| Ar | 20.39 ± 4.62 | 2.98 ± 1.82 |

Note:
$r_{xy}$, Mean relatedness for individual pairs of diploid individuals; Afs, average number of full-siblings per individual; Asor, average number of second order relatives per individual; Ator, average number of third order relatives per individual; Ar, average number of relatives (full-sibling/parent-offspring relationships, second-order relationships and third-order relationships).

The highest positive correlation between the different relatedness estimators can be observed between the TrioML and DyadML estimators, which further confirms that the relatedness estimates provided by the TrioML estimator best describes the relatedness of the birds in the given sample (Table S4, Fig. S1). Furthermore, TrioML exhibited positive and statistically significant correlations with the other estimators (Table S5), further confirming the proposed relatedness among the samples. Relatedness estimates of 95% CI TrioML were used to reconstruct the family tree which is presented in Fig. S2, and the dyads that demonstrated the full-sibling or parent-offspring relationship are presented in Fig. 1. In order to assess if the obtained relationship can be categorized as a full-sibling or parent-offspring relationship, we have used the ML-Relate software. This software suggested that the dyads of interest exhibit the following relationship: S31/SF1—full-siblings, S95/S53—full-siblings (same nest), SA3/S01—full-siblings, BE423/S34—parent-offspring and S033/S43—full-siblings.

To further evaluate the maternal relationship between the proposed full-siblings we have analysed the mtDNA of the birds from these dyads. The resulting haplotypes are presented in Table 4. Birds in the dyads S31/SF1 and S95/S53 possess the same mtDNA haplotypes which suggest that the birds in dyads have direct maternal ancestor which supports the full-sibling relationship. For the dyads SA3/S01 and S033/S43 the direct maternal ancestor can be excluded since their mtDNA haplotypes are different. The sequencing of the mtDNA regions for the BE423 sample showed that there is no direct maternal link thus negating the parent-offspring relationship in BE423/S34 dyad. The observed discrepancy between the mtDNA and microsatellite data could be the result of the marker choice or specific demographic history of the Griffon vulture population from Serbia that underwent a strong bottleneck event at the end of the 20[th] century.

## DISCUSSION

Assessing the sex ratio in the population is an important factor in determining the strength and health of a population and is of great importance considering the managing strategies for small and endangered populations (*Ghorpade et al., 2012*; *Griffiths & Tiwari, 1995*;

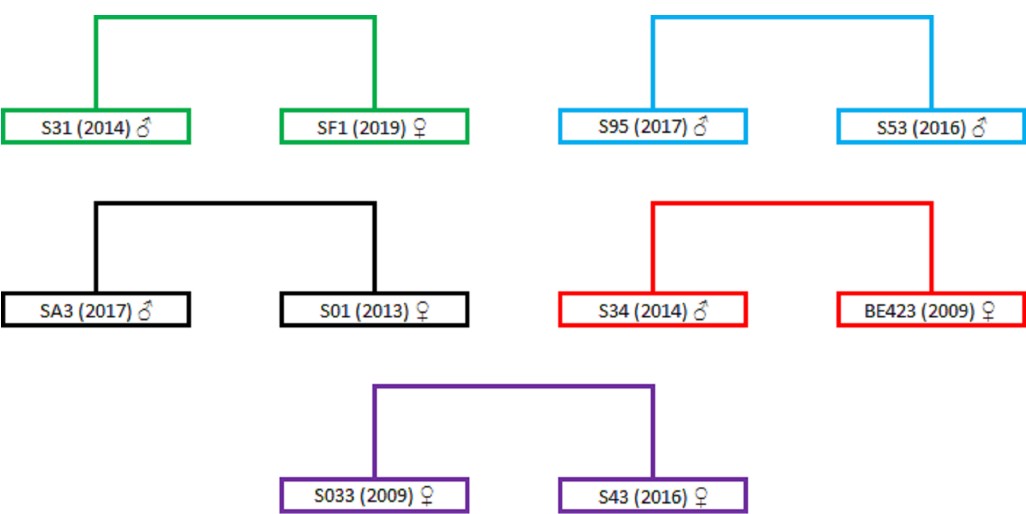

**Figure 1 Dyads representing full-sibling or parent-offspring relationships.** Dyads are in different colors representing the relationship between two different birds. Individual birds are represented with the ring markings that they have received during the annual monitoring activity (S31, SF1, S95, S53, SA3, S01, S34, BE423, S033 and S43); numbers in the brackets represent the year of hatching/marking; sex is represented using the following symbols: ♂ for males and ♀ for females. For better presentation the dyads were extracted from Fig. S2.

**Table 4 MtDNA haplotypes of the birds from the sample of the Griffon vulture population from Serbia that according to the COANCESTRY analysis exhibited the full-sib/parent-offspring relationship.**

| Sample ID | Dyad | *Cytb* haplotype | *D-loop I* haplotype |
|---|---|---|---|
| S31 | S31/SF1 | 13884C 13887C 13896T 14650C | 15188G 15752T 15791A 15830T |
| SF1 | S31/SF1 | 13884C 13887C 13896T 14650C | 15188G 15752T 15791A 15830T |
| S95 | S95/S53 | 13884C 13887C 13896T 14650C | 15188G 15752T |
| S53 | S95/S53 | 13884C 13887C 13896T 14650C | 15188G 15752T |
| SA3 | SA3/S01 | 14650C | 15188G 15618G 15752T 15830T |
| S01 | SA3/S01 | 13884C 13887C 13896T 14650C | 15188G 15752T |
| S34 | S34/BE423 | 13884C 13887C 13896T 14650C | 15188G 15752T 15791A 15830T |
| BE423 | S34/BE423 | 13884C 13887C 13896T 14650C | 15188G 15343G 15346T |
| S033 | S033/S43 | 13884C 13887C 13896T 14650C 14682G | 15188G 15343G 15346T 15451A 15752T 15791A 15830T |
| S43 | S033/S43 | 14650C | 15188G 15752T 15830T |

**Note:**
MtDNA haplotypes are determined by the analysis of *Cytb* and *D-loop I* regions and comparison to the reference mitogenome NC_036050.1.

*Lambertucci et al., 2013*; *Robertson et al., 2006*). In monogamous species, the optimal sex ratio among adult birds that maximizes population growth is 1:1 (*Legendre, 2004*) and it is expected to be in equilibrium at birth (*Cockburn, Legge & Double, 2002*). In our sample of the Griffon vulture population from Serbia, we didn't detect a statistically significant deviation from the expected sex ratio of 1:1. The balanced sex ratio in Griffon vulture populations were also detected in the native population in Spain (*Wink et al., 1998*) and populations in France, both native and reintroduced, (*Bosé et al., 2007*), while in Italy the sex balance in the reintroduced population was shifted towards males (*Garofalo et al.,*

*2016*). The skewed sex ratio in Italy could be the result of external factors such as the sex-biased stock of founders used for reintroduction. It was recognized as an alarm pointing to the importance of proper restocking strategies and the necessity to maintain sex balance in the groups of released birds as well as performing constant monitoring of small and endangered populations (*Garofalo et al., 2016*). Maintaining a balanced sex ratio was also shown to be important for population overall fitness in other monogamous and sexually monomorphic species of *Gyps* genus such as the Oriental White-backed Vulture (*Gyps bengalensis*) or Long-billed Vulture (*Gyps indicus*) (*Arshad, Chaudhary & Wink, 2009*) and distant groups of monogamous and sexually monomorphic species like King penguins (*Aptenodytes patagonicus*) (*Bordier et al., 2014*). In our sample, over the course of 9 years, we have detected 2 years with skewed sex balance, although not statistically significant. In the year 2013, the balance shifted towards males while the year 2020 demonstrated the shift towards females. Strong and viable populations that inhabit a healthy environment with enough resources should exhibit a balanced sex ratio (*Fisher, 1930*) while it was shown that in the case of the poor condition of parents or deprived resources parents tend to invest in less costly sex (*Brommer et al., 2003*; *Chakarov et al., 2015*; *Ferrer, Newton & Pandolfi, 2009*; *Rosenfield et al., 2015*; *Trivers & Willard, 1973*). The study of pollutant influence in the environment, like mercury contamination, on the formation of sex in birds demonstrated the skewed balance toward females among fledglings in three different bird species where females are considered as less costly sex: belted kingfisher (*Megaceryle alcyon*), eastern bluebird (*Sialia sialis*) and tree swallow (*Tachycineta bicolor*) (*Bouland et al., 2012*). This negative influence on overall health further confirmed the hypothesis that the parents with poorer health tend to favour less costly sex among their offspring, the females (*Bouland et al., 2012*). On the other hand, in raptors, where sexual dimorphism is present, females are larger than males making males the less costly sex in terms of parental care investment (*Pleasants, 1988*; *Trivers & Willard, 1973*). Although the Griffon vulture belongs to the raptor group, they are sexually monomorphic, meaning there is no observable difference in the cost of breeding males or females. The skewed balance in 2013 could be explained either by the stochasticity during the sampling process or by the influence of the environment that made the parents invest more in males. It has been shown that in some circumstances when birds have greater access to food or are in a better condition they tend to produce more males (*Bradbury & Blakey, 1998*; *Sutherland, 2002*; *Whittingham & Dunn, 2000*). The skewed balance in 2020 may be contributed to the fact that four birds that fell from the nests during an incident in the Special nature reserve "Uvac" were included in the sampling. The incident included the illegal flyover of the helicopter which forced a number of birds to prematurely jump from their nests into the lake. Among the four birds that were saved from drowning three were females. Overall, the presented 9-year data show no deviation from the balanced sex ratio among offspring which further supports the findings that the Griffon vulture population of Serbia is strong and viable and strengthen its prospects to be used as a stock population for further reintroduction efforts in the Balkan Peninsula (*Davidovic et al., 2020*, *2022*; *Marinkovic et al., 2021*).

Calculated absolute relatedness variation among relatedness estimators for our dataset is expected since not all of them take into account inbreeding and/or genotyping errors and have different mathematical methods for sample size accommodation and allele weighing (*Taylor, 2015*). Due to its lowest variance and the fact that it takes inbreeding into account, TrioML was found to be the most reliable estimator for our data but other estimators may be more suitable in other circumstances (*Csillery et al., 2006*; *Hogg et al., 2018*; *Rollins et al., 2012*; *Taylor, 2015*). According to the TrioML estimator, each bird in our sample has around 20 relatives and around three relatives according to 95% CI for the same estimator suggesting that percentage of related birds is around 5.3%. This value is comparable to the one found in other birds of prey like Aplomado falcon (*Falco femoralis*) whose values ranged from 5.2–8.8% (*Johnson et al., 2021*). In addition, the detected level of relatedness exhibits nearly four times lower values of relatedness compared to the values found in endangered Egyptian vulture (*Neophron percnopterus*) in Spain (*Blanco & Morinha, 2021*). Further evaluation of the mtDNA lineages of the most closely related dyads showed the importance of the proper choice of relatedness estimator and the need to include other markers in the analysis. While *Cytb* and *D-loop I* sequences corroborated full-sibling relationship for our test birds from the same nest (dyad S95/S53) and another pair originating from different nests (dyad S31/SF1), the other two proposed full-sibling dyads (SA3/S01 and S033/S43) have different mtDNA haplotypes negating the full-sibling status. The proposed parent-offspring relationship in BE423/S34 dyad also can be rejected since not only do these birds have different mtDNA haplotypes but records showed that the proposed mother was rescued and in recovery in the Palić zoo in the year the proposed son was hatched and marked. Discrepancies in the full-sibling dyad may be contributed to the fact that although the full-siblings will share 50% of their alleles on average more variance from dyad to dyad is expected, making related individuals closely related than in reality due to the choice of the marker (*Weir, Anderson & Hepler, 2006*). It also may suggest an inbreeding signature on neutral genetic variability and bottleneck effect. Most of the now living birds in the Serbian Griffon vulture population are descendants from 10 breeding pairs in the 1990s and it is expected that some birds may share a higher proportion of the same alleles due to the stochasticity of a bottleneck. This is further corroborated by the mean $F$ which is 0.08268, indicating that the parents of the average individual in the Serbian population are more closely related than outbred first cousins whose offspring by definition have $F = 0.0625$ (*Nietlisbach et al., 2017*). Although it seems high, the inbreeding level of 8% detected for the Griffon vulture population of Serbia is not unusual compared to other analysed populations in Europe. The same level of inbreeding was detected in the population from Israel, while the population from Cyprus exhibited a lower value, $F = 0.07$ (*Arshad et al., 2009*). The lowest level of inbreeding was detected in the Griffon vulture population from Spain ($F = 0.01$) which is expected since this population is the biggest and the most stable population in Europe and does not have a history of dramatic population decline (*Arshad et al., 2009*). Furthermore, when compared with the inbreeding levels of the related species with similar biology and behaviour, like the African Cape vulture (*Gyps coprotheres*), the $F$ value is well in the range found in different populations of this species (−0.5 to 0.19) (*Kleinhans & Willows-Munro, 2019*). In the same study, the inbreeding levels

were analysed for the South African populations of bearded vultures (*Gypaetus barbatus*), $F = 0.17$, hooded vultures (*Necrosyrtesmonachus*), $F = -0.07$, and African White-backed vulture (*Gyps africanus*), $F = 0.07$ (*Kleinhans & Willows-Munro, 2019*). Other related species such as Oriental White-backed Vulture and Long-billed Vulture exhibited inbreeding levels of 0.01 and 0.07 respectively (*Arshad et al., 2009*). The same inbreeding levels may be attributed to the philopatric nature of the vultures as well as detected historical population declines recorded in most of the analysed species and populations. What clearly distinguishes the Serbian Griffon vulture population from other analysed ones is the fact that it retains a high level of microsatellite diversity comparable to ones in abundant populations (*Davidovic et al., 2020*).

## CONCLUSIONS

Results presented in this study further support the evidence that the Serbian Griffon vulture population is stable, healthy and has a great perspective to be used as a stock population for further conservation efforts in the Balkan Peninsula from the genetic diversity point of view. Even though a relatively high level of $F$ was detected among the birds from the Serbian Griffon vulture population, the detected values are in the range of values found in other Griffon vulture populations and related species. In addition, previous research showed comparable values of the genetic diversity on the level of microsatellites and *Cytb* between the Griffon vulture population from Serbia and other populations with the identification of private alleles (*Davidovic et al., 2020*) and haplotypes (*Davidovic et al., 2022*) present only in the Balkan Peninsula. In order to preserve the specific genetic diversity of the Griffon vulture population from the Balkan Peninsula, the future restocking efforts in this region should use the birds originating from the autochthonous Balkan population.

## ACKNOWLEDGEMENTS

We want to thank the Birds of Prey Protection Foundation for making the collection of the samples possible by organizing the monitoring, logistics, organization of the alpinists who climb into the nests, providing the accommodation during the fieldwork and providing the tags for the birds. We also want to thank the management and rangers of the "Special Nature Reserve Uvac" and speleologists from different speleological clubs who helped us during the fieldwork. We thank the reviewers of this manuscript for their constructive and friendly comments which helped us in improving this article.

### Funding

This work was funded by the Ministry of Education, Science and Technological Development of the Republic of Serbia (Grant Nos. 451-03-68/2022-14/200007 for Slobodan Davidović, Saša Marinković, Aleksandra Patenković, and Marija Tanasković; 451-03-9/2021-14/200178 for Marina Stamenković-Radak). The Birds of Prey Protection Foundation provided the fees for the alpinists and financially supported the necessary

material and logistics for the fieldwork. The funders had no role in study design, data collection and analysis, decision to publish, or preparation of the manuscript.

## Grant Disclosures

The following grant information was disclosed by the authors:
Ministry of Education, Science and Technological Development of the Republic of Serbia: 451-03-68/2022-14/200007 and 451-03-9/2021-14/200178.

## Competing Interests

The authors declare that they have no competing interests.

## Author Contributions

- Slobodan Davidović conceived and designed the experiments, performed the experiments, analyzed the data, prepared figures and/or tables, authored or reviewed drafts of the article, and approved the final draft.
- Saša Marinković conceived and designed the experiments, authored or reviewed drafts of the article, provided the samples, and approved the final draft.
- Irena Hribšek performed the experiments, analyzed the data, authored or reviewed drafts of the article, provided the samples, and approved the final draft.
- Aleksandra Patenković performed the experiments, prepared figures and/or tables, authored or reviewed drafts of the article, and approved the final draft.
- Marina Stamenković-Radak conceived and designed the experiments, authored or reviewed drafts of the article, and approved the final draft.
- Marija Tanasković conceived and designed the experiments, authored or reviewed drafts of the article, and approved the final draft.

## Animal Ethics

The following information was supplied relating to ethical approvals (*i.e.*, approving body and any reference numbers):

The Ministry of Nature Protection of the Republic of Serbia, Ministry of Agriculture, Forestry and Water Management of the Republic of Serbia (323-07-09135/2020-05/1) approved the study.

## Field Study Permissions

The following information was supplied relating to field study approvals (*i.e.*, approving body and any reference numbers):

The ethical committee of the Institute for Biological Research "Siniša Stanković", Belgrade (323-09135-2020-05) approved the study.

## DNA Deposition

The following information was supplied regarding the deposition of DNA sequences:

The data is available at GenBank: OL962644, OL962685, OL962664, OL962652, OL962667, OL962636, OL962648, OL962634, OL962646, OL962649 and ON988042–ON988051.

## Data Availability

The raw data is available in the Supplemental Files and GenBank: OL962644, OL962685, OL962664, OL962652, OL962667, OL962636, OL962648, OL962634, OL962646, OL962649 and ON988042–ON988051.

## Supplemental Information

Supplemental information for this article can be found online at http://dx.doi.org/10.7717/peerj.14477#supplemental-information.

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
