# Peer review of "Sex ratio and relatedness in the Griffon vulture (Gyps fulvus) population of Serbia"

_PeerJ, doi:10.7717/peerj.14477_

## Round 0.1 · original submission · Major Revisions

The research work carried by Slobodan et al. is interesting in the area of population genetics. The study is ambitious for the conservation of Griffon vultures.

Reviewer 1 ·

Basic reporting

See below

Experimental design

See below

Validity of the findings

See below

Additional comments

General comment:
The sex ratio represents a determining success factor in a population that shows a stable monogamous reproductive system such as the griffon vulture. An alteration of this gender relationship assumes a decisive value in populations of limited size of pairs. Another not exactly positive aspect of an alteration in the sex ratio and inbreeding could also derive from the low exchange of individuals with other populations. Knowledge of the sex ratio also plays a fundamental role in restocking programs aimed not only to maintain the species in a territory but also for the conservation and dissemination of genetically features in order to maintain a wide biodiversity within the Gyps fulvus. Therefore the study of the population of Serbia has an important value for the purpose of maintaining biodiversity.
Here are some comments and suggestions sorted by lines that could contribute to improve the paper:
Introduction
lines 72-75 on the causes that have brought the Serbian griffon population to only 10 pairs and also which factors led to the increase from 10 to 164 pairs such as farm feeding station or immigration;

line 79 add mtDNA after mitochondrial DNA

lines 90-94 to make known whether there is a limited exchange of individuals between the Serbian population with the neighboring populations of Italy, Greece, Bulgaria and Croatia;

lines 95 - 117 relating to the sex ratio which is also reported in the discussion with similar detail, please rewrite this part limiting only to the essential;

lines 121-125; it is important to know the sex ratio at the nest rather than as an adult which could be influenced by external factors;

Material
line 162 from 2013-2021;

line 164 please indicate the vein of sampling;

lines 166-168 several data on the chicks were reported but we do not know at what age the blood collection was made, please you could indicate a range period;

lines 173-178 it is not clear if the 57 unrelated birds from 56 nests are additional samples or just a part included in the 169 samples. Please, clarify;

line 195 homology is a qualitative parameter referring to a common origin. In this case I would suggest to rephrese in a similar way: “P2 reverse primer 5’ TCTGCATCGCTAAATCCTTT 3’ complementary to the CHD 3’ region of both W and Z chromosomes;

line 211 remove the dot after 2018;

line 213 remove mitochondrial and brachets;

lines 226-231 why did the Authors use different pairs of primer for amplification and sequencing of the D-loop I region? Did they use an internal primer pair to get better results in sequencing? If yes, the length of the D-loop I sequence should be shorter than the size of the corresponding PCR fragment. Please, clarify this point;

Results
line 269 delete the dots after the years and over the years you have had the opportunity to do the blood collection from the same pairs. In this way you could know the gender alternation by year. if these data have been acquired why not let them know;

lines 270-271 delete the dot after 2014 and rephrase as follows: The number of hatched chicks per year was in constant increase with the exception of the years 2014 and 2021;

lines 276-277 delete the dots after 2019 and Table 1;

line 279 delete the dot after 2020. Please carefully check the rest of the main text and remove the dots when not needed;

line 345 did the Authors mean that the sequencing of the mtDNA regions didn’t show any matrilinear relationship? If yes, please replace amplification with sequencing;

lines 343-346 even if you mention this in the discussion, can you provide a brief explanation for the missing overlap between the SSR and mtDNA results from the SA3/S01, S033/S43 and BE423/S43 dyads? It is not mandatory but IMO it could be helpful for the reader;

Discussion
Lines 360-364 the unbalanced sex ratio shifted towards males in the Italian population is related to external factors please enter this aspect

line 374 remove the article “the” before poor;

lines 377-380 by reading the first part of the sentence I understand that, in such a condition, the sex in birds is unbalanced toward females, while following with the reading I get the opposite meaning. I would suggest revising and rephrasing the whole sentence to better clarify this point;

line 384 remove the article “the” before terms;

lines 387-389 why should the environmental condition lead the parents to invest more in males? It could be useful for non-expert readers;

lines 430-31 the level of significance of genetic biodiversity cannot be compared between the griffon vulture populations of Cyprus and Israel that have undergone the restocking action to that of Serbia which is totally a local population.

·

Basic reporting

1. There are inconsistencies present within this article. For example, the use of analysis and analyze is found in line 301. Please stick to one spelling. Furthermore, line 370 mentions a period of seven years whereas in the rest of the article the period of study is defined as eight years. Line 162 states the period to be 2013-202 instead of 2013-2021.

2. The English used is not as professional as it should be. The use of the word “like” in lines 77, 117, 129, 402 and 442 can be changed for similar or comparable. Please make sure date periods are written in full rather than ‘90s use 1990s (lines 73, 155 and 423). There should not be a full stop after any years unless it is the end of the sentence (see lines 74, 173-175, 211, 269-279).

3. There are sections in the article where the language is ambiguous and should be reworded in clear English. See lines 63-65, 83-85, 120-121, 142-147, 153-157, 161-175, 250-252, 253, 296-302.

4. The aims of the study are not clear and should be redefined. Example “In this paper, we evaluated the sex ratio and estimated the relatedness among Griffon vulture offspring present in Serbia. The aims of this study were to first assess the impact of the bottleneck event that occurred in the early 1990s and second to investigate the potential of using this population as a stock population for reintroduction in the Balkan Peninsula.”

5. Line 161-162 reword this sentence and include locality coordinates as there is no graph to show the region.

6. Line 174-175 is not clear. Are the recaptured samples relevant to the study period? Please clarify.

7. Move lines 175-178 to the molecular analyse section.

8. Delete line 183

9. Move lines 204-208 to 197. It should read “Amplification was performed in a volume of 25 µL with the following final concentrations of reaction components: 1×Taq Buffer with (NH4)2SO4, 2.5mM MgCl2, 0.8 mM dNTP mix, 1 U of reverse Taq polymerase (all components were produced by Thermo Fisher Scientific, EU) and 10 pmol of each forward and reverse primer. All PCRs were performed on the 2720 Thermal Cycler205 (Applied Biosystems, UK) using the following cycling conditions…”

10. Move line 209 to line 200. It should read “elongation was performed at 72 °C for 10 min. Amplification fragments were evaluated by agarose gel electrophoresis.”

11. Define the accession number in line 239.

12. Line 253-254 mentions “same parameter settings” to define the same as what? Be specific.

13. Line 254-260 is clumsy please use symbols <, >, ≤,≥.

14. Line 306 delete “The chosen estimator” and use TrioML.

15. The article is lacking background on the importance of vultures in the ecosystem. Why have the authors used the Griffon vulture as a model? This vulture species is of least concern, state why research should focus on such species before they reach endangered status. Why is this species of ecological importance? You mentioned a bottleneck event but why should these birds be reintroduced? Mention the decline rates in this area and the importance of preserving this species.

16. Please reference the work see line 385.

17. Please be consistent with spacing example F=0.27 and F =0.27.

18. Table and figure legends need to be more descriptive and labelled correctly. For example, “Table 1: Ratio of hatched and sampled chicks with sex ratio through the years” includes the number of chicks, state the year period 2013-2021 and state specie Griffon vulture. The titles and legends should be so descriptive that they can stand alone, and the reader will understand the content.

19. The raw data is shared. Please make sure that the microsatellite alleles are in integers Supp Table S1. Please describe Supp Table S1 b. What do the numbers in the table represent?

20. Supp Table S4 should be added to the main article and not supplementary. The authors used seven parameters but did not report and describe all of them in the text. What are their significances?

21. Figure 1 should be redone as a Table and clearly described.

22. Table 2 should include the number of birds sampled each year.

23. There are few data sets varying in sample sizes example 1596 (Table S4), 1182 and 169 (Figure 1) and 57 (microsatellite data and relatedness). Please clarify and clearly define the differences in each Table and Figure as well as the body of text.

24. For 2013*- including the birds captured in 2009 and 2011. What is the significance of including birds captured in 2009 and 2011? Should you also include 2013-2021 excluding birds captured in 2009 and 2011 and compare if it skews the data?

Experimental design

1. This article requires a more clearly defined research question. See point 15 under “Basic reporting”. More information is required as to why this study is important.

2. Although this article included microsatellite data more analysis could have been conducted using this data, for example including parentage analyses. Especially as relationships were supposed due to the assumption that one pair of Griffon vultures uses the same nest throughout their reproductive period. This can easily be checked using the microsatellite data.

Validity of the findings

This paper highlights the significance of sex ratio and relatedness analyses in reintroduction programs. The authors conduct a multiple genetic marker approach (including microsatellite loci, mtDNA markers and sexing CHD primers) in analysing a subsect of Griffon vulture, Gyps fulvus in Serbia. This study is meaningful and will add value to the literature. However, the conclusion that foreign vulture populations should not be used as a source population for restocking needs more clarification. In addition, more analysis can be conducted using the microsatellite data.

---

## Round 0.2 · Minor Revisions

The manuscript has been significantly improved. The concerns raised by reviewer 2 need to be properly addressed.

Reviewer 1 ·

Basic reporting

ok

Experimental design

ok

Validity of the findings

ok

·

Basic reporting

1. Ln 57 change , It to , it

2. Ln 61-64 change to the following: "Although once common species in Europe, these vultures faced serious population decline in the mid-20th century leading to the extinction of many populations across the European range and vast depletion of breeding birds across the breeding range/globally.”

3. Ln 65-75 are clumsy rewrite and spelling errors. “Thanks to the successful cross-continental conservation efforts that included reintroduction and conservation measures implemented across the former natural range of the Griffon vulture in Europe, the future of this ecologically important species seems brighter REFERENCES. The European populations are now showing a significant increment in size with this species now being categorized as ‘Least Concern’ on the IUCN Red List. REFERENCES (Dobrev et al. 2021; Le Gouar et al. 2008; Pirastru et al. 2021; Potena et al. 2009). However, even with these successful measures, this species is still facing serious population declines in some habitats and has become extinct in others, which is why it is a subject of conservation strategies by numerous laws, directives, and conventions (reviewed in Pirastru 2021).

4. Ln 79-83 reword as: Due to conservation efforts that included the opening of the feeding station and prohibition of poison usage, the Serbian Griffon vulture population is one of the successful conservation stories. From only 10 breeding pairs recorded in the 1990s, it has reached 164 breeding pairs in 2019 and now is the largest population in the Balkan Peninsula (Dobrev et al. 2021; Marinkovic et al. 2021).

5. Ln 83-87 are clumsy and confusing reword. The historical bottleneck, seen in the last decade of the 20th century, was a peak of a rapid demographic decline in Serbian Griffon vulture populations. This decline resulted from mass poisonings and loss of food sources due to the change in livestock management (Davidovic et al. 2020).

6. Ln 87-91 reword. A study by Davidovic et al. (2020) showed that this population's genetic variability (based on the variability of 10 microsatellites loci) was similar to numerous, unscathed populations such as the Iberian Griffon vulture population and therefore is proposed to serve as a stock population for further reintroduction efforts in the Balkan Peninsula.

7. Ln 102 remove duplicate text. Serbian population of Griffon vulture population

8. Ln 137 reword as affects opposite sexes differently.

9. Ln 138 reword. Traditional sex determination of individual adult Griffon vulture is…

10. Ln 140-141reword. Thus, molecular sexing was developed to replace these invasive and difficult fieldwork methods

11. Ln 152-156 are repetitive reword

12. Ln 158-160 reword. ‘…pedigree data, traditional markers of choice are microsatellites, usually a panel of eight - 30 microsatellite loci

13. Ln 165-166 reword. The program COANCESTRY (Wang 2011) reflects population reality because it takes inbreeding into account

14. Ln 167 full stop after events

15. Ln 167-171. COANCESTRY estimates two types of identification of related individuals. The first estimate is relatedness (r) which is a continuous measure of the proportions of the alleles of the dyad (pair of individuals) that are IBD relative to a reference population. The second type infers kinship which are discrete categories of dyad relationships (parent-offspring, full-sibling, half-sibling).

16. Ln 196 add the in “Among the 169 blood..”

17. Ln 208-211 do not fit under DNA extraction

18. Ln 223 program change to programme be consistent.

19. Ln 226 rewrite as All amplified fragments were

20. Ln 235 change was to were

21. Ln 238-240 if you doing to format primers like this be consistent look at lines 216-219

22. Ln 258 add the “can be accessed by the following”

23. Ln 265 you use ten but lines 81, 87 you use 10 be consistent

24. Ln 274 change was to were

25. Rewrite lines 277-282 as. The threshold values of rxy coefficient were as follows: rxy < 0.09375 the birds in the dyad were unrelated, 0.09375  rxy < 0.1875 were considered as third-order relatives (relationship), 0.1875  rxy < 0.375 were considered as second-order relatives (relationship) and rxy  0.375

26. Ln 294 delete second full stop

27. Ln 297-299 rewrite as Some of the nests were visited multiple times during this period. The chicks originating from the supposed same breeding pair of birds that were hatched in different years were sampled.

28. Ln 301- 302 rewrite clumsy one of the sexes was more prominent than the other

29. Ln 314 change full stop to comma 2013, although

30. Ln 327 delete second full stop

31. Ln 331 rewrite as previous work of Davidovic et al. (2020)

32. Ln 335 delete (Wang 2011)

33. Ln 343 be consistent with sibs and siblings

34. Ln 345 delete each

35. Ln 358 change describe to describes

36. Ln 391 change was to were

37. Ln 395 delete ie and use such as

38. Ln 400 italicise Gyps

39. Ln 401- 403 change like to such as

40. Ln 446 moderate in comparison to what? Give an example in the literature.

41. Ln 449-450 rewrite as (dyads S95/S53 and S31/SF1)

Experimental design

No comment

Validity of the findings

No comment

Additional comments

This article is much improved. There are a few discrepancies in the text and grammatical errors that I have highlighted in basic reporting.

---

## Round 0.3 · accepted · Accept

I am happy to accept the manuscript for publication in PeerJ.